# Contingency management for tobacco smoking during opioid addiction treatment: a randomised pilot study

Tom Stephen Ainscough,[1,2] Leonie S Brose,[1,2] John Strang,[2] Ann McNeill[1,2]

► Prepublication history and additional material are available. To view these files please visit the journal online (http://dx.doi.org/10.1136/bmjopen-2017-017467).

## ABSTRACT

**Introduction** Smoking rates among individuals in treatment for opioid addiction are close to five times that of the general public. Moreover, drug-addicted smokers have a premature mortality rate four times greater than drug-addicted non-smokers. The aim of this pilot study was to investigate whether contingency management (CM) can be successfully added to evidence-based stop smoking treatment in individuals undergoing treatment for opioid addiction and assess preliminary evidence for its impact.

**Participants** Forty tobacco smokers currently undergoing treatment for opioid addiction.

**Intervention** Escalating with reset CM as an adjunct to standard smoking cessation treatment. Financial incentives will be administered over a 5-week period for either biochemically verified abstinence from smoking or attendance at the clinic. Participants will be randomised to conditions stratified on current levels of smoking (high or low).

**Objectives and analyses** To assess whether a CM intervention can be successfully added to standard stop smoking services treatment, in patients undergoing outpatient treatment for opioid addiction. This will be measured as the number of people completing the 5 weeks of the intervention.

**Ethics and dissemination** Ethics approval for the study was granted on the 16 June 2016 by the London—city and east (reference 16/LO/0990) ethics committee. The pilot study was retrospectively registered on clincaltrials. gov in January 2017 (ID: NCT03015597). A SPIRIT checklist and figure are available for this protocol. It is planned that the results of this study will be published in an academic journal.

¹UK Centre for Tobacco and Alcohol Studies, UK
²IoPPN, King's College London, London, UK

**Correspondence to**
Mr Tom Stephen Ainscough;
thomas.ainscough@kcl.ac.uk

**Strengths and limitations of this study**

► Extends an extensively tested evidence-based intervention to a novel treatment population.
► Implements a randomised controlled experimental design.
► Due to constraints of the intervention, blinding of both participants and treatment centre staff is not possible.

## BACKGROUND

Tobacco smoking is the leading cause of premature death in the Western world,[1] currently killing six million people per year across the globe, predicted to rise to eight million people annually by 2030.[2] In England alone, smoking killed 74 000 people in 2014.[3] Consequently, tobacco smoking places a large economic burden on both the National Health Service and the larger UK economy. It has been estimated that tobacco smoking costs the NHS approximately two billion pounds per year, with a total cost to the UK economy of approximately 13 billion pounds annually.[4]

In 2016, smoking prevalence in the general UK population fell below 17% for the first time.[5] However, despite this encouraging downwards trend, smoking prevalence among those in treatment for drug addiction remains high, with a prevalence of 88% recorded in the UK in 2013[6] and little change observed in the 20 years from 1988 to 2008.[7] Drug-addicted smokers also have a fourfold greater premature mortality rate than non-smokers.[8] This situation is further exacerbated by evidence showing that the efficacy of the standard stop smoking treatment currently used is nearly halved when an individual has used illicit drugs in the past 30 days.[9] There is, therefore, a great need for the development of novel interventions for tobacco smoking for those in drug addiction treatment that can bolster the efficacy of current interventions. One of the highest rates of smoking prevalence in substance abuse treatment is observed in opioid addictions treatment, ranging between 84% and 98%.[7 10–13] Moreover, those in treatment for opioid addiction report high rates of interest in stop smoking treatment,[10 11] making them an ideal population for the development of interventions for tobacco smoking in substance abuse treatment.

Contingency management (CM) is a behavioural intervention based on the principles of operant conditioning, whereby

**BMJ**

changes in behaviour are brought about by positively rewarding desired behaviours. CM has been shown to be an effective intervention for drug use during opioid addiction[14] and has been recommended for use in opioid addictions in the UK for some time.[15] Some studies show promising results for CM in smoking cessation during treatment for opioid addiction[16–20]; however, this remains under-researched. Moreover, none of the currently published studies investigating this were carried out in the UK, or alongside standard stop smoking treatment.

The aim of the proposed pilot study was to assess whether a CM intervention can be successfully added to standard stop smoking services treatment in patients undergoing outpatient treatment for opioid addiction.

## ETHICS
### Risks to participants
There is no known risk associated with the CM behavioural intervention. Smoking cessation can precipitate a number of uncomfortable withdrawal symptoms. These will be attenuated by the stop smoking services treatment provided at the treatment centre, an evidence-based treatment that includes nicotine replacement therapy (NRT), e-cigarettes and behavioural support. Any information recorded from participants will be anonymised using a participant ID number, the master sheet for which will be stored in a locked cabinet at the treatment centre. This ensures that no identifiable information will ever leave the treatment centre.

### Vouchers rather than cash
The treatment centre where the pilot study is being carried out did not want participants to be paid in cash so as not be able to buy cigarettes, alcohol or drugs. The 'Love2Shop' vouchers used as an alternative can be spent in a number of high street stores. Although cash vouchers have been shown to be more effective than vouchers in some case,[21] other research has shown cash and monetary vouchers to be of equal efficacy.[22 23] The use of monetary vouchers, therefore, should not negatively impinge on the efficacy of the current intervention. Participants will receive both the study intervention and standard stop smoking services treatment at no cost.

## METHODS/DESIGN
This protocol was designed in accordance with the SPIRIT (Standard Protocol Items: Recommendations for Interventional Trials) statement. See online supplementary material for SPIRIT checklist and online supplementary figure.

### Objectives
Primary objective: To investigate whether a CM intervention can be successfully added to standard stop smoking services treatment, in patients undergoing outpatient treatment for opioid addiction, in order to identify any elements that need changing before carrying out a full-scale randomised controlled trial (RCT).

Secondary objective: To gather preliminary findings regarding the effects of the CM intervention on smoking in this group, and any possible effects the intervention may have on opioid addiction treatment outcomes.

### Participants, recruitment, inclusion criteria and randomisation
As this is a pilot study, the primary outcome is not the efficacy of the study intervention. Consequently, the sample size has not been calculated to ascertain efficacy. Instead, the method outlined by Viechtbauer et al[24] for calculating the sample size based on the probability of any issues that may arise has been used. A sample size of 40 using the above rationale is powerful enough to provide over 90% certainty of detecting any issues that occur with a probability of over 5%.

The study therefore aims to recruit 40 patients, all undergoing current treatment for opioid addiction and who smoke ten or more cigarettes a day. Participants will be recruited from the study site, an outpatient drug addiction treatment centre, either through self-referrals in response to advertisements shown in the treatment centre or referrals from treatment centre staff. Participants are eligible for inclusion if they want to quit smoking (complete abstinence), are aged between 18 and 65 years, undergoing pharmacological treatment for opioid addiction, smoke a minimum of 10 cigarettes per day and provide informed consent. Use of smoking cessation medication is not a criterion for exclusion. Participants will be ineligible for inclusion in the study if they exhibit insufficient English skills to understand study protocols, are currently undergoing treatment for other drugs of abuse or if taking part in other research. Pregnant women will not be excluded.

Participants will be randomised into either experimental (CM for abstinence) or control (CM for attendance) conditions when recruited into the trial. Randomisation will be performed by the principal investigator (PI), using the service provided by the company 'sealed envelope',[25] and will be performed using random permuted blocks within strata. Randomisation will be stratified based on participants' current smoking frequency (between 10 and 20 per day, and more than 20 per day[6]). All participants will be given at least 24 hours after being given an information sheet to decide whether to take part, and will provide written consent, collected by the PI (TSA).

### Study design
A two-arm randomised controlled pilot study with 6-month follow-up. The intervention will be provided as an adjunct to the standard smoking cessation treatment provided at the treatment centre, with CM rewards available during 2–5 weeks of the smoking cessation treatment. The study will be conducted in compliance with the principles of the Declaration of Helsinki,[26] the principles of Good Clinical Practice and all applicable regulatory requirements.

## Opioid treatment

As part of the standard opioid treatment programme, the clinic offers both behavioural and pharmacological treatments. Pharmacological treatments include methadone, buprenorphine and in some cases a combination of buprenorphine and naloxone; each of these progresses from a daily supervised dose, to a daily unsupervised pickup to a weekly unsupervised pickup. All medication prescriptions are reviewed every 6 months. Clients are also allocated a key worker with whom they meet in person every 2 weeks to discuss their treatment, and who can refer them to a number of different behavioural support programmes. These include psychological therapies or group therapy for their drug use, or a number of other services for issues related to their drug use such as needle exchanges, bloodborne virus testing and domestic violence support. In the past, the clinic has implemented CM interventions as part of other research projects; however, CM has never been implemented as part of the standard opioid treatment programme.

## Standard treatment

Prior to the initiation of this study, the smoking clinic had not operated for several months; smoking cessation training was therefore readministered to clinic staff and the smoking cessation treatment relaunched prior to the start of the trial. The treatment runs at the same time each week, on a Monday afternoon between 2 and 4 PM. The standard smoking cessation treatment provided at the treatment centre follows the treatment programme set out by the National Centre for Smoking Cessation and Training (NCSCT)[27] and The National Institute for Health and Care Excellence (NICE) guidelines for smoking cessation.[28] This treatment combines manualised behavioural support to stop smoking with NRT and takes place over 6 weeks with one session per week. In the context of drug addiction treatment, service users are sometimes offered treatment over a slightly longer period of time. In the first meeting, the service user's readiness and ability to quit is assessed, information for the remainder of the treatment programme is given and a quit date for the next week is set. For the remaining 5 weeks, clients attend the clinic to receive behavioural support and have their abstinence biochemically verified. In the study clinic, NRT is available free of charge to all individuals engaged with smoking cessation treatment, in the form of nicotine patches, gum, inhalators, mouth or oral spray and oral strips. At the time of the study, the clinic is also additionally offering (on a trial basis) e-cigarettes, which have a nicotine content of 18 mg/ml. These e-cigarettes are disposable and securely sealed, initially designed for use in high-security environments such as prisons.[29] The smoking cessation treatment provided at the treatment centre does not include treatment with bupropion.

During the 6 weeks of treatment, service users are given a week's supply of NRT or e-cigarettes at a time. At the end of the 6 weeks, service users are given a further 2-week supply of NRT or e-cigarettes before exiting the treatment. The type of NRT received is decided by clients with guidance from the cessation worker, and can constitute a single form of NRT or a combination of different types. Clients' breath carbon monoxide (CO) levels are measured using a Bedfont piCO+ Smokerlyzer breath CO monitor. Measurements are taken at the initial visit and at each subsequent visit over the next 5 weeks, to biochemically verify self-reported abstinence from smoking (CO<10 ppm[30]). NRT and e-cigarette use is recorded throughout treatment. Participants are made aware of these procedures in the participant information sheet that they are given prior to signing consent to the study (see online supplementary appendix 1).

## CM intervention

The CM intervention will run as an adjunct to the normal smoking cessation treatment, and follows an escalating with reset schedule. In escalating with reset CM, rewards increase in a set increment value for each successive verified display of the desired behaviour. When the desired behaviour is not observed, no reward is given, and the reward value for the next verified display of the desired behaviour is reset to that of the initial reward. Reward values then begin to rise again in the same way as before. The CM intervention will run for 5 weeks in total, starting in week 2 of the standard stop smoking services treatment and ending in week 6 (table 1). Randomisation will be performed after collection of demographics following taking of consent. Participants will be rewarded for smoking abstinence in the experimental condition, or for attending the smoking cessation clinic in the control condition. Smoking abstinence will be defined as a breath CO reading of <10 ppm, and attendance will be defined as attending the smoking cessation treatment at the

| Table 1 Reward schedule | | | | | | |
|---|---|---|---|---|---|---|
| Smoking cessation treatment week No | 1 | 2 | 3 | 4 | 5 | 6 |
| CM week No | | 1 | 2 | 3 | 4 | 5 |
| Reward value | £0.00 | £5.00 | £10.00 | £20.00 | £40.00 | £40.00 |

Reward schedule for a participant that remains abstinent and/or attends all smoking cessation treatment meetings (dependent on condition) for the duration of the intervention. Maximum total reward: £115.
CM, contingency management.

clinic that week. After each smoking cessation treatment session, the cessation worker will fill out a slip that records each participant's individual participant number and his or her breath CO for that day. The cessation worker will give these slips to the PI who will sit in an adjacent room and administer rewards where appropriate. All participant data will be recorded using participant numbers ensuring that no identifiable data leave the clinic, and will be stored in an encrypted file, separate to a sheet matching participant names to IDs which will be kept in a locked office at the treatment centre. Due to the nature of the CM intervention, it is not possible to blind participants to treatment allocation. Cessation workers will not be made aware of treatment allocation; however, they cannot be considered to be blinded to treatment allocation as it is possible that clients may discuss this with them.

Reward values will be the same in both conditions and begin at £5, doubling each time the incentivised behaviour is recorded to a maximum of £40. All rewards will be given as 'Love2Shop' vouchers. Over the course of the whole intervention, participants will be able to earn a maximum of £115 (table 1). At the end of the CM intervention, participants will be asked to complete a client satisfaction and well-being survey, which was previously used to assess client satisfaction of stop smoking services treatment.[31]

## Measures

### Outcome measures

The primary outcome will be assessed by recording the number of participants completing the 5 weeks of the intervention in each condition. Success will be defined as 60% or more of participants completing treatment.

The secondary objectives of the study are to gather preliminary findings regarding the effects of the CM intervention on smoking in this group, and any possible effects the intervention may have on opioid addiction treatment outcomes. Smoking abstinence will be recorded as point prevalence and biochemically verified with abstinence defined as a breath CO reading of under 10 ppm[30]. Participants were informed that smoking cannabis would increase CO levels.

Participant medical records will be accessed after completion of the intervention to ascertain participants' opioid addiction treatment, including treatment adherence, drug types (methadone, Subutex, so on), dosage and schedule (daily supervised pickup, weekly pickup, so on) as well as illicit drug use throughout the period of the trial.

### Follow-up measures

At the 6-month follow-up (see below for follow-up procedures), the following measures will be recorded:

Point prevalence smoking abstinence: Self-reported smoking abstinence for 7-days before follow-up and exhaled air CO<10 ppm.[30]

Continuous abstinence: Self-reported smoking abstinence since end of treatment and exhaled air CO<10 ppm. Participants smoking five or fewer cigarettes during the 6-month follow-up will be considered self-reported quitters.[30]

Illicit drug use, collected at the end of the study from participants' medical records.

All those lost to follow-up will be treated as smoking.[30]

### Other measures

At the first stop smoking treatment session, a number of demographic and smoking behaviour variables will be recorded. The collection form for this information is shown in online supplementary appendix 2. As many contact details as possible will also be taken for the participants in order to increase the probability of participants being able to be followed-up. This will include the details of relevant friends and family members. Participants will also complete a satisfaction questionnaire on the last day of their participation in the trial, which will assess a number of satisfaction criteria including the value of incentives received (see online supplementary appendix 3).

### Follow-up procedures

Six months after their set quit date, participants will be contacted by the PI to ascertain their self-reported smoking status. The main purpose of this follow-up is to ascertain whether participants can be successfully followed-up for 6 months, and no group differences are expected to be found between the different conditions. To test the optimal follow-up method, participants will be pseudo-randomised by recruitment order to be contacted by text and phone call, or email and phone call. All participants will also be asked to return to the clinic in order to have their breath CO levels tested to verify abstinence. Once this is done, participants will have completed their participation in the study. Participants will receive a £10 voucher for completing the follow-up procedure.

### Planned analysis

As the primary objective of the intervention does not entail any hypothesis testing, the only statistics reported for this will be descriptive, namely means and SD for the number of participants retained at the end of treatment in each condition. Baseline demographics will be compared between conditions using t-tests for continuous and $\chi^2$ test for categorical data to ensure that any differences in these are not driving any potential differences in retention.

For the secondary objectives, differences between the groups in smoking cessation will be investigated using $\chi^2$ test, differences between conditions on opioid use and opioid treatment during the intervention will be compared using t-tests and $\chi^2$ tests dependent on data and any questionnaire data will be reported using descriptive

statistics. All statistics will be performed as two-tailed tests using an alpha value of 0.05.

## DISCUSSION

The addition of contingent incentives to standard evidence-based smoking cessation treatment in opiate addiction clients will be an innovative approach, having never been attempted before in the UK.

The current trial has a number of limitations that should be improved in future studies. First, the value and frequency of rewards in this study are comparatively lower than those of previous trials and should, therefore, be increased to encourage the cessation. The use of breath CO only in measuring abstinence is not the most rigorous method available for testing, due to the relatively short period of time it takes for breath CO levels to return to levels considered as those of a non-smoker. Urine cotinine levels provide a more rigorous measure of abstinence; however, they are confounded by the use of NRT, therefore necessitating the measurement of anabasine instead. The measurements of both cotinine and anabasine were beyond the scope of the current intervention. Furthermore, provision of incentives to participants in the attendance group should come before breath CO levels are measures to avoid the risk of these participants thinking their incentives are linked to CO levels.

However, the intervention has a number of potential strengths. If feasible, the intervention will be easily disseminated, and it has the potential to be an effective intervention for smoking in this client group. Pilot studies are an imperative step in the development of complex interventions, and form the first step on the road to full-scale RCT and potentially implementation.[32][33] If successful, this programme paves the way for the development of a full-scale RCT of CM for smoking in opiate addiction treatment, which would include an economic evaluation, and potential trials for smokers in other drug addiction treatment.

**Contributors** TSA was responsible for the design of the study with input from AM, LSB and JS. TSA is responsible for the recruitment of participants and for the collection, and analysis of participant data with input from AM and LSB.

**Funding** This work was funded as part of TSA's PhD studentship by the Medical Research Council and the Institute of Psychiatry, Psychology and Neuroscience (MRC/IoP Excellence Studentship). LSB is funded by a Cancer Research UK (CRUK)/ BUPA Foundation Cancer Prevention Fellowship (C52999/ A19748). LSB and AM are members of the UK Centre for Tobacco and Alcohol Studies, a UK Clinical Research Collaboration Public Health Research: Centre of Excellence. Funding from the Medical Research Council, British Heart Foundation, Cancer Research UK, Economic and Social Research Council and the National Institute for Health Research under the auspices of the UK Clinical Research Collaboration is gratefully acknowledged 35 (MR/K/K023195/1). Neither the funding bodies nor study sponsors had any role in study design; collection, management, analysis and interpretation of data; writing of the report and the decision to submit the report for publication.

**Competing interests** JS has contributed to UK guidelines which include consideration of the potential role of contingency management in the management of addiction problems (NICE, 2007; chaired by JS), and JS also chaired the broader-scope pan-UK working group preparing the 2007 and 2017 editions of the 'Orange Book' ('Guidelines on the Management of Drug Misuse & Dependence') for the UK Departments of Health, providing guidance on management and treatment of drug dependence and misuse, which include guidance on possible inclusion of contingency management. JS's institution has received support and funding from the Department of Health (England) and National Treatment Agency (England), and JS and JS's institution have provided funded consultancy advice on possible novel addiction treatments, products and formulations to a range of pharmaceutical companies but these do not have any connection to the intervention being investigated in this paper. JS's employer (King's College London) has registered intellectual property on a novel buccal naloxone with which JS is involved, and JS has been named in a patent registration by a pharmaceutical company as inventor of a potential novel concentrated nasal spray, but these do not have any connection to the work being reported in this paper. A fuller account of JS's interests is at http://www.kcl.ac.uk/ioppn/depts/addictions/people/hod.aspx.JS is supported by the National Institute for Health Research (NIHR) Biomedical Research Centre for Mental Health at South London and Maudsley NHS Foundation Trust and King's College London and is an NIHR Senior Investigator.

**Ethics approval** London—city and east.

**Provenance and peer review** Not commissioned; externally peer reviewed.

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
