## [Reviewer comments · BMJ Open]

ARTICLE DETAILS

TITLE (PROVISIONAL)	Contingency management for tobacco smoking during opioid addiction treatment: a randomised pilot study
AUTHORS	Ainscough, Tom; Brose, Leonie; Strang, John; McNeill, Ann

VERSION 1 - REVIEW

REVIEWER	Bryan Hartzler University of Washington, US
REVIEW RETURNED	11-May-2017

GENERAL COMMENTS	This is an unmasked review of a manuscript describing design of a pilot trial to test the efficacy of contingency management as adjunct to standard smoking cessation services among persons undergoing outpatient opiate treatment. The manuscript outlines a pilot trial design wherein a proposed 40 adult tobacco smokers undergoing outpatient opiate treatment will participate in a six-week period to a tobacco cessation program (involving manualized behavioral support and nicotine replacement therapy) with randomization to an experimental contingency management condition that reinforces tobacco abstinence during the program or a control condition that reinforces program attendance. Stated trial objectives are to examine if a CM intervention can be feasibly added to the tobacco cessation program, and to gather preliminary data concerning CM effects on tobacco use. The trial design appears well-conceived and appropriate for this timely, often overlooked substance challenge of opiate treatment patients. While rather brief, the manuscript does offer a compelling rationale for the trial and its objectives, descriptions of its regulatory oversight, participant recruitment/consenting, reinforcement strategy and procedures, treatment conditions, outcome measurement, and planned analyses. My lone suggestion is that, given the stated plan to access participant medical records to examine potential illicit substance use during the study period, the authors consider additional secondary outcomes concerning opiate treatment adherence (i.e., agonist medication dosing, clinic visits) that may be ascertained from these records. Such information may reveal broader health benefits of the tobacco cessation program and/or adjunctive use contingency management to then merit examination as secondary outcomes in a full-scale RCT.
---

REVIEWER	Gavin Bart Hennepin County Medical Center, USA
REVIEW RETURNED	15-May-2017

GENERAL COMMENTS	This protocol submission lays out a pilot study for what is assumed to be a future study of smoking cessation treatment for patients receiving outpatient treatment for opiate addiction. The authors note that the primary objective is to assess whether a contingency management intervention can be successfully added to standard smoking cessation treatment in patients receiving treatment for opiate addiction. The writing of the primary objective with the use of the term “successfully added” reads like an implementation study where the plan would be to identify how well the intervention is adopted by the clinic, yet there is no evaluation of the implementation within the proposal, rather this is a standard clinical trial pilot study. It would be more appropriate to restate the primary outcome as the effect of the intervention on smoking outcome such as self-report point abstinence and biologically confirmed maker. Some minor comments are added below: It may be appropriate to consider changing opiate to opioid throughout the manuscript as this allows a more encompassing patient population. There is no description of the opioid addiction treatment offered by the clinic: methadone, buprenorphine, naltrexone, behavioral, etc. Given that the clinic already offers smoking cessation treatment, have those wanting this treatment already been exhausted and that the remaining population is refractory or uninterested in cessation and thus not generalizable? Is CM used elsewhere in the clinic for opiate addiction treatment outcome?
---

REVIEWER	Kelly Dunn, Ph.D. Johns Hopkins University United States
REVIEW RETURNED	22-May-2017

GENERAL COMMENTS	This is an interesting and exciting pilot study protocol proposal. The study will be the first to evaluate contingency management + pharmacotherapy for smoking cessation with opiate-treatment patients in the UK. The study is balancing rigor and feasibility and will use these data to decide whether a subsequent RCT evaluation of this intervention is warranted. The authors included several standard features of CM in this protocol. I have direct experience in this area and below have made some suggestions that I believe are worth considering at this stage in study development in order to increase the likelihood the investigators will detect an effect of CM in this challenging group. Vouchers:  • Cash vouchers have not been uniformly shown to be more effective than non-cash vouchers (see Vandrey et al., PMC2043576;
---

Festinger et al., PMC4504189), so the investigators should feel confident in their decision to deliver non-monetary incentives for abstinence in this study.

Inclusion:

- Is there a specific time frame in which people will need to want to stop smoking (e.g., within the next 30 days?). Otherwise this could be problematic, as many people report they want to quit smoking but are unable to settle on a target quit date.
- Will the investigators restrict the population to individuals receiving a specific kind of pharmacotherapy in this initial pilot study? Evidence suggests that opioid agonists increase the reinforcing effects of cigarettes, and there may be differences in the reinforcing effects among patients receiving full-agonists (methadone) vs. partial agonists (buprenorphine). Since this is a small pilot/feasibility that is looking for whether a signal is present, the authors may want to reduce variability related to pharmacotherapy.
- Will participants who are currently attempting to quit smoking using methods such as bupropion, varenicline, or NRT still be eligible for study inclusion?
- Since CO is the primary outcome, the investigators may want to restrict active cannabis smokers, regularly collect self-report of cannabis smoking, and/or inform participants that any smoking (including cannabis) will increase CO levels to preserve the relationship between tobacco smoking and CO levels.

CM Intervention:

- It is CRITICAL that the participants understand that the attendance incentive is truly independent of their smoking status, and the current design (wherein participants in both groups receive the incentive after their visit) will undermine that contingency. I would suggest that the attendance incentive be provided to patients IMMEDIATELY upon beginning the session and BEFORE collection of the CO sample, to keep those behaviors distinct. Otherwise the participants will falsely come to believe that their incentive is being at least partly based upon their smoking status – which will undermine your control group.
- Will participants be attending the clinic once weekly to earn incentives? Though incentive resets have been shown empirically to strengthen CM interventions, you must also balance that with participant interest in the study. If a participant provides a positive sample and earns nothing and knows that his/her earning the following week will also be substantially reduced- it may prevent him/her from attempting to reinitiate abstinence – especially since those two weeks would represent 40% of possible earnings. Resets are valuable when there is a high frequency of visits, but in this case (where there are only 5 possible incentives) the authors may want to not pay participants for positive samples but allow them to earn the payment that had been withheld at the following visit to encourage another quit attempt.
- A breath CO of ≤ 10 ppm is not very sensitive to smoking behavior and can be achieved relatively easily following several hours of abstinence (for instance- smokers who provide a CO in the morning after sleeping for several hours can easily make this threshold). Several recent studies recommend setting CO cutoffs for smoking cessation to ≤ 3 ppm (Raiff et al., PMC2910872; Cropsey et al., PMC4207872; Emery et al., PMID: 26386471); the authors should consider reducing their CO threshold. You can also consider requiring afternoon visits to ensure daytime abstinence
- Urinary cotinine testing would provide a more rigorous method of

	measuring abstinence in this design since you will only be testing participants once weekly- however the NRT in this study will confound any cotinine testing. If the authors decide at some point to switch to another form of pharmacotherapy, urinary cotinine would be a good outcome measure for this once-weekly visit design. Outcomes:  • Since the investigators are not conducting an efficacy evaluation, it seems unnecessary to evaluate outcomes at 6-months. It may be sufficient to evaluate 30 or 90-day outcomes and then use those data to power and/or support a subsequent RCT that would evaluate outcomes at longer time-periods (like 6 months). No existing study of CM with pharmacotherapy for smoking cessation in opiate-treatment patients has reported sustained effects of the intervention 6-months after the CM protocol was completed and those studies were much more intensive than the one proposed here. Thus, proposing a 6-month follow-up adds quite a bit of complexity and time to study completion with only a very low likelihood that group differences will be observable. • What are the opiate addiction treatment outcomes that will be evaluated? Are the authors referring to ongoing illicit drug use, retention in opiate treatment, attendance at counseling visits, etc... With this small a sample and low intensity study, it seems unlikely that large effects will be observed. • Also consider taking detailed notes regarding adherence to NRT to guide the design of any subsequent trials and enable analyses regarding the extent to which NRT may have impacted outcomes. • In general, I think the outcomes could be better operationalized and that specific a priori analyses could be proposed. Questionnaires:  • Unclear on what type of scale the “quitting confidence”, “importance”, and “readiness” outcomes will be collected. • Smoking behaviour: I assume the authors will use the conventional “0-30 min, >30min” cutoff that has been highly associated with nicotine dependence severity? • May want to ask participants to rate the relative value of the incentives and the NRT in their quit attempt as an additional measure of participant satisfaction with treatment, in support of a subsequent trial. References  • This is the most recent evaluation of CM for smoking cessation in opiate-treatment patients and should be referenced here: Sigmon et al., PMC4826799.
--	---

VERSION 1 – AUTHOR RESPONSE

Reviewer: 1

Reviewer Name: Bryan Hartzler

Institution and Country: University of Washington, US

Please state any competing interests: None declared.

My lone suggestion is that, given the stated plan to access participant medical records to examine potential illicit substance use during the study period, the authors consider additional secondary outcomes concerning opiate treatment adherence (i.e., agonist medication dosing, clinic visits) that may be ascertained from these records. Such information may reveal broader health benefits of the tobacco cessation program and/or adjunctive use contingency management to then merit examination

as secondary outcomes in a full-scale RCT.

- We acknowledge that the use of participant data for secondary outcomes was less well specified than it could be. This has now been changed from:

“Participant medical records will be accessed to ascertain participants’ opioid addiction treatment, including drug types (methadone, Subutex etc.), and dosage, as well as illicit drug use throughout the period of the trial.”

To now read:

“Participant medical records will be accessed after completion of the intervention to ascertain participants’ opioid addiction treatment, including treatment adherence, drug types (methadone, Subutex etc.), dosage and schedule (daily supervised pickup, weekly pickup etc.) as well as illicit drug use throughout the period of the trial.”

Reviewer: 2

Reviewer Name: Gavin Bart

Institution and Country: Hennepin County Medical Center, USA

Please state any competing interests: None declared

It may be appropriate to consider changing opiate to opioid throughout the manuscript as this allows a more encompassing patient population.

- The term opiate has been changed for opioid throughout the document.

There is no description of the opioid addiction treatment offered by the clinic: methadone, buprenorphine, naltrexone, behavioural, etc.

- The following section has now been added to address this:

“Opioid Treatment

As part of the standard opioid treatment programme the clinic offers both behavioural and pharmacological treatments. Pharmacological treatments include methadone, buprenorphine and in some cases a combination of buprenorphine and naloxone; each of these progresses from a daily supervised dose, to a daily unsupervised pickup to a weekly unsupervised pickup. All medication prescriptions are reviewed every six months. Clients are also allocated a key worker with whom they meet in person every two weeks to discuss their treatment, and who can refer them to a number of different behavioural support programs. These include psychological therapies or group therapy for their drug use, or a number of other services for issues related to their drug use such as needle exchanges, blood-borne virus testing and domestic violence support. In the past, the clinic has implemented CM interventions as part of other research projects, however CM has never been implemented as part of the standard opioid treatment program.”

Given that the clinic already offers smoking cessation treatment, have those wanting this treatment already been exhausted and that the remaining population is refractory or uninterested in cessation and thus not generalizable?

- This is a not concern for the current trial as the smoking cessation treatment that once ran at the study centre had broken down. The following section has been added to the beginning of the Standard Treatment section to reflect this:

“Prior to the initiation of the current study, the smoking clinic had not operated for several months; smoking cessation training was re-administered to clinic staff and the smoking cessation treatment re-launched prior to the start of the trial.”

Is CM used elsewhere in the clinic for opiate addiction treatment outcome?

- The clinic has implemented CM interventions in the past for other drug use but has never used CM in the treatment of opioid addiction. A new section describing the standard opioid addiction treatment program has been added to incorporate this:

“Opioid Treatment

As part of the standard opioid treatment programme the clinic offers both behavioural and pharmacological treatments. Pharmacological treatments include methadone, buprenorphine and in some cases a combination of buprenorphine and naloxone; each of these progresses from a daily supervised dose, to a daily unsupervised pickup to a weekly unsupervised pickup. All medication prescriptions are reviewed every six months. Clients are also allocated a key worker with whom they meet in person every two weeks to discuss their treatment, and who can refer them to a number of different behavioural support programs. These include psychological therapies or group therapy for their drug use, or a number of other services for issues related to their drug use such as needle exchanges, blood-borne virus testing and domestic violence support. In the past, the clinic has implemented CM interventions as part of other research projects, however CM has never been implemented as part of the standard opioid treatment program.”

Reviewer: 3

Reviewer Name: Kelly Dunn, Ph.D.

Institution and Country: Johns Hopkins University, United States

Please state any competing interests: No conflicts of interest

Vouchers:

Cash vouchers have not been uniformly shown to be more effective than non-cash vouchers (see Vandrey et al., PMC2043576; Festinger et al., PMC4504189), so the investigators should feel confident in their decision to deliver non-monetary incentives for abstinence in this study.

- This section has been re-written to now read:

“The treatment centre where the pilot study is being carried out did not want participants to be paid in cash so as not be able to buy cigarettes, alcohol, or drugs. The “Love2Shop” vouchers used as an alternative can be spent in a number of high street stores. Although cash vouchers have been shown to be more effective than vouchers in some cases [1], other research has shown cash and non-monetary vouchers to be of equal efficacy [2,3]. The use of monetary vouchers therefore should not negatively impinge on the efficacy of the current intervention. Participants will receive both the study intervention and standard stop smoking services treatment at no cost.”

Inclusion:

Is there a specific time frame in which people will need to want to stop smoking (e.g., within the next 30 days?). Otherwise this could be problematic, as many people report they want to quit smoking but are unable to settle on a target quit date.

- The information sheet that participants receive makes it clear that they will have to set a quit date exactly 7 days after their initial study visit. This is standard practice for all smoking clinics following the NCSCT (National Centre for Smoking Cessation and Training) treatment program. The participant information sheet has been added to the appendix and the following section has been added to the end of the standard treatment section:

“Participants are made aware of these procedures in the participant information sheet that they are given prior to signing consent to the study (see appendix 2).”

Will the investigators restrict the population to individuals receiving a specific kind of pharmacotherapy in this initial pilot study? Evidence suggests that opioid agonists increase the reinforcing effects of cigarettes, and there may be differences in the reinforcing effects among patients receiving full-agonists (methadone) vs. partial agonists (buprenorphine). Since this is a small pilot/feasibility that is looking for whether a signal is present, the authors may want to reduce variability related to pharmacotherapy.

- For the purposes of the current pilot study, the primary interest was the ability to add a contingency

management intervention to an existing smoking cessation treatment, therefore participants undergoing all types of opioid addiction treatment were eligible to participate.

Will participants who are currently attempting to quit smoking using methods such as bupropion, varenicline, or NRT still be eligible for study inclusion?

- As part of the current protocol, participants were eligible for inclusion regardless of current quit attempts so long as they were smoking 10 or more cigarettes per day. Participants were however asked about any past quit attempts including any use of NRT, varenicline or e-cigarettes (see appendix 1. Bupropion is not available at the research clinic. The following sentence has been added to the Participants, recruitment, inclusion criteria and randomisation section to reflect this: "Use of smoking cessation medication is not a criterion for exclusion."

Since CO is the primary outcome, the investigators may want to restrict active cannabis smokers, regularly collect self-report of cannabis smoking, and/or inform participants that any smoking (including cannabis) will increase CO levels to preserve the relationship between tobacco smoking and CO levels.

- The primary concern of the current pilot study was to assess how well a contingency management intervention could be added to standard stop smoking services treatment. In future pilot studies of this intervention however, it would be important to be mindful of this potential problem. Participants were made aware during the intervention that smoking cannabis would increase their CO levels. A sentence has been added to the outcome measures section: "Participants were informed that smoking cannabis would increase CO levels."

CM Intervention:

It is CRITICAL that the participants understand that the attendance incentive is truly independent of their smoking status, and the current design (wherein participants in both groups receive the incentive after their visit) will undermine that contingency. I would suggest that the attendance incentive be provided to patients IMMEDIATELY upon beginning the session and BEFORE collection of the CO sample, to keep those behaviors distinct. Otherwise the participants will falsely come to believe that their incentive is being at least partly based upon their smoking status – which will undermine your control group.

- We thank the reviewer for alerting us to this and fully agree. Unfortunately, as the trial is already under way, we are unable to change the administration of the incentive at this point. For any future trials, we will certainly take this into account. We have added a limitations section to the discussion to ensure dissemination of these considerations:

"The current trial has a number of limitations that should be improved upon in future studies. Firstly, the value and frequency of rewards in the current study are comparatively lower than those of previous trials. If the primary objective of the intervention was abstinence, both the reward value and frequency should be increased to encourage cessation. The use of breath CO only in measuring abstinence is not the most rigorous method available of testing abstinence, due to the relatively short period of time it takes for breath CO levels to return to levels considered as those of a non-smoker. Urine cotinine levels provide a more rigorous measure of abstinence, however are confounded by the use of NRT therefore necessitating the measurement of anabasine instead. The measurement of both cotinine and anabasine were beyond the scope of the current intervention. Furthermore, provision of incentives to participants in the attendance group should come before breath CO levels are measures to avoid the risk of these participants thinking their incentives are linked to CO levels."

Will participants be attending the clinic once weekly to earn incentives? Though incentive resets have been shown empirically to strengthen CM interventions, you must also balance that with participant interest in the study. If a participant provides a positive sample and earns nothing and knows that his/her earning the following week will also be substantially reduced- it may prevent him/her from

attempting to reinitiate abstinence – especially since those two weeks would represent 40% of possible earnings. Resets are valuable when there is a high frequency of visits, but in this case (where there are only 5 possible incentives) the authors may want to not pay participants for positive samples but allow them to earn the payment that had been withheld at the following visit to encourage another quit attempt.

- We agree with this but again are unable to change this for the present trial at this point. Incentive values are low in any case and we have added this to the new limitations section.

A breath CO of • While breath CO levels have been discussed we are using the standard outcome measure for smoking cessation in the UK. This is based both on the Russell Standard [4], and NICE (National Institute For Health and Clinical Excellence) quality standard guidelines [5]. CO levels were recorded at each visit. We have added to the protocol that all visits were in the afternoon:

“The treatment runs at the same time each week, on a Monday afternoon from 2-4 PM.”

Urinary cotinine testing would provide a more rigorous method of measuring abstinence in this design since you will only be testing participants once weekly- however the NRT in this study will confound any cotinine testing. If the authors decide at some point to switch to another form of pharmacotherapy, urinary cotinine would be a good outcome measure for this once-weekly visit design.

- As above, we have added this suggestion to the new limitations and future trials section.

“The current trial has a number of limitations that should be improved upon in future studies. Firstly, the value and frequency of rewards in the current study are comparatively lower than those of previous trials and should therefore be increased to encourage the cessation. The use of breath CO only in measuring abstinence is not the most rigorous method available of testing abstinence, due to the relatively short period of time it takes for breath CO levels to return to levels considered as those of a non-smoker. Urine cotinine levels provide a more rigorous measure of abstinence, however are confounded by the use of NRT therefore necessitating the measurement of anabasine instead. The measurement of both cotinine and anabasine were beyond the scope of the current intervention. Furthermore, provision of incentives to participants in the attendance group should come before breath CO levels are measures to avoid the risk of these participants thinking their incentives are linked to CO levels.”

Outcomes:

Since the investigators are not conducting an efficacy evaluation, it seems unnecessary to evaluate outcomes at 6-months. It may be sufficient to evaluate 30 or 90-day outcomes and then use those data to power and/or support a subsequent RCT that would evaluate outcomes at longer time-periods (like 6 months). No existing study of CM with pharmacotherapy for smoking cessation in opiate-treatment patients has reported sustained effects of the intervention 6-months after the CM protocol was completed and those studies were much more intensive than the one proposed here. Thus, proposing a 6-month follow-up adds quite a bit of complexity and time to study completion with only a very low likelihood that group differences will be observable.

- We are not expecting to find group differences at the six-month outcome but are instead interested in assessing whether a long-term follow-up can successfully be conducted in this patient group. The following has been added to the protocol under the follow-up heading to highlight this fact:

“The main purpose of this follow-up is to ascertain whether participants can be successfully followed up for six months, and no group differences are expected to be found between the different conditions.”

What are the opiate addiction treatment outcomes that will be evaluated? Are the authors referring to ongoing illicit drug use, retention in opiate treatment, attendance at counseling visits, etc... With this

small a sample and low intensity study, it seems unlikely that large effects will be observed.

- We have added a little more information about these outcomes. As for the previous comment, because this is a pilot trial, we are not necessarily expecting to find effects.

“Participant medical records will be accessed after completion of the intervention to ascertain participants’ opioid addiction treatment, including treatment adherence, drug types (methadone, Subutex etc.), dosage and schedule (daily supervised pickup, weekly pickup etc.) as well as illicit drug use throughout the period of the trial. “

Also consider taking detailed notes regarding adherence to NRT to guide the design of any subsequent trials and enable analyses regarding the extent to which NRT may have impacted outcomes.

- Details of the NRT prescribed to participants during the course of the intervention are recorded, however adherence to NRT was not recorded due to the nature of the trial’s primary objective.

In general, I think the outcomes could be better operationalized and that specific a priori analyses could be proposed.

- We agree that the proposed outcomes are not as fully formed as it would be for a full trial, however as this is the first time that an intervention of this nature has been formed in the UK, the current trial is necessarily more exploratory in nature. The following sentence has been added to the planned analysis section to better outline the analysis of secondary outcomes:

“differences between conditions on opioid use and opioid treatment during the intervention will be compared”

Questionnaires:

Unclear on what type of scale the “quitting confidence”, “importance”, and “readiness” outcomes will be collected.

- The available options for all of the questions asked to participants during the baseline assessment have now been added to the appendix.

Smoking behaviour: I assume the authors will use the conventional “0-30 min, >30min” cutoff that has been highly associated with nicotine dependence severity?

- These cut-offs have not been used, and instead the cut-offs recommended in the NCSCT (National Centre for Smoking Cessation and Training) guidelines have been used instead, as these are the cut-offs currently used as standard practice in smoking cessation services in England. These still allow the data collected to be broken into these groupings however. The options available for participants on this question have been added to the appendix.

May want to ask participants to rate the relative value of the incentives and the NRT in their quit attempt as an additional measure of participant satisfaction with treatment, in support of a subsequent trial.

- Participants were asked these questions and a number of others, in a post intervention satisfaction survey, a copy of which has been added to the appendix.

References

This is the most recent evaluation of CM for smoking cessation in opiate-treatment patients and should be referenced here: Sigmon et al., PMC4826799.

- Thank you, we have added this reference to the introduction.

References

1 Topp L, Islam MM, Day CA. Relative efficacy of cash versus vouchers in engaging opioid

substitution treatment clients in survey-based research. J. Med. Ethics. 2013;39:253–6.

2 Vandrey R, Bigelow GE, Stitzer ML. Contingency management in cocaine abusers: a dose-effect comparison of goods-based versus cash-based incentives. Exp Clin Psychopharmacol 2007;15:338–43. doi:10.1037/1064-1297.15.4.338

3 Festinger DS, Dugosh KL, Kirby KC, et al. Contingency management for cocaine treatment: cash vs. vouchers. J Subst Abuse Treat 2014;47:168–74. doi:10.1016/j.jsat.2014.03.001

4 West R, Hajek P, Stead L, et al. Outcome criteria in smoking cessation trials: proposal for a common standard. Addiction 2005;100:299–303. doi:10.1111/j.1360-0443.2004.00995.x

5 National Institute For Health and Clinical Excellence. Smoking: supporting people to stop | Guidance and guidelines | NICE. <https://www.nice.org.uk/guidance/qs43/chapter/Quality-statement-5-Outcome-measurement> (accessed 6 Jun 2017).

VERSION 2 – REVIEW

REVIEWER	Bryan Hartzler University of Washington, US
REVIEW RETURNED	14-Jun-2017

GENERAL COMMENTS	The authors have adequately addressed the lone concern raised in my original review of this trial design manuscript. Further, they appear to have made reasonable effort to address, where possible, concerns of the other reviewers.
---

REVIEWER	Gavin Bart, MD PhD Hennepin County Medical Center USA
REVIEW RETURNED	16-Jun-2017

GENERAL COMMENTS	The revised manuscript addresses previous referee concerns. There are no additional comments at this time.
--

REVIEWER	Kelly Dunn Johns Hopkins University School of Medicine
REVIEW RETURNED	28-Jun-2017

GENERAL COMMENTS	I am unfamiliar with this particular publication type and did not realize it was meant to describe an existing study. Many of my comments were meant to be helpful to the researchers before they undertook the study, and were not as relevant to a protocol that was already underway. Nevertheless, the authors were very responsive to my comments, and I believe the manuscript is strong and describes the study well. This is an important but under-researched area, I wish the researchers luck with their protocol.
---